# Cotton (*Gossypium hirsutum*) VIRMA as an N6-Methyladenosine RNA Methylation Regulator Participates in Controlling Chloroplast-Dependent and Independent Leaf Development

**DOI:** 10.3390/ijms23179887

**Published:** 2022-08-31

**Authors:** Xiaoyu Huang, Nigara Abuduwaili, Xinting Wang, Miao Tao, Xiaoqian Wang, Gengqing Huang

**Affiliations:** 1Hubei Key Laboratory of Genetic Regulation and Integrative Biology, School of Life Sciences, Central China Normal University, Wuhan 430079, China; 2Xinjiang Key Laboratory of Special Species Conservation and Regulatory Biology, College of Life Science, Xinjiang Normal University, Urumuqi 830054, China

**Keywords:** cotton, leaf development, GhVIR, m^6^A modification, chloroplast biosynthesis

## Abstract

N6-methyladenosine (m^6^A) is one of the most abundant internal modifications of mRNA, which plays important roles in gene expression regulation, and plant growth and development. Vir-like m^6^A methyltransferase associated (VIRMA) serves as a scaffold for bridging the catalytic core components of the m^6^A methyltransferase complex. The role of VIRMA in regulating leaf development and its related mechanisms have not been reported. Here, we identified and characterized two upland cotton (*Gossypium hirsutum*) VIRMA genes, named as *GhVIR-A* and *GhVIR-D*, which share 98.5% identity with each other. *GhVIR-A* and *GhVIR-D* were ubiquitously expressed in different tissues and relatively higher expressed in leaves and main stem apexes (MSA). Knocking down the expression of *GhVIR* genes by the virus-induced gene silencing (VIGS) system influences leaf cell size, cell shape, and total cell numbers, thereby determining cotton leaf morphogenesis. The dot-blot assay and colorimetric experiment showed the ratio of m^6^A to A in mRNA is lower in leaves of *GhVIR-VIGS* plants compared with control plants. Messenger RNA (mRNA) high-throughput sequencing (RNA-seq) and a qRT-PCR experiment showed that GhVIRs regulate leaf development through influencing expression of some transcription factor genes, tubulin genes, and chloroplast genes including photosystem, carbon fixation, and ribosome assembly. Chloroplast structure, chlorophyll content, and photosynthetic efficiency were changed and unsuitable for leaf growth and development in *GhVIR-VIGS* plants compared with control plants. Taken together, our results demonstrate GhVIRs function in cotton leaf development by chloroplast dependent and independent pathways.

## 1. Introduction

To date, more than 170 RNA modifications have been identified; N6-methyladenosine (m^6^A) is the most abundant internal modification of mRNA (0.1~0.4% m^6^A/A), which corresponds to an average of 3~5 m^6^A per mRNA [1,2,3]. Studies have shown that mRNA m^6^A modification widely exists in yeast, plants, flies, mammals, as well as viruses and is a highly conserved methylation modification [3,4]. M^6^A modification in mRNA influences numerous fundamental cell processes and acts as a key switch on mRNA metabolism, including mRNA transcription, degradation, translation efficiency, alternative polyadenylation, secondary structure, nuclear retention/export, and splicing [3,4,5,6,7]. In plants, m^6^A has been shown to affect various plant developmental processes [3,7,8]. Accumulating evidence has indicated that mRNA m^6^A modification plays crucial roles in the development of embryo and shoot apical meristems, leaf morphogenesis, root growth, trichome branching, floral transition, and early callus induction [7,8]. In addition, it is also shown that m^6^A modification regulates the land plant mitochondria and their putative effects on organellar gene expression [9]. However, the significance and cellular role of m^6^A RNA modification in organelles remain largely unknown.

In mammals, mRNA m^6^A methylation is catalyzed by the methyltransferase complex, which is a 200 kDa multi-subunit protein complex consisting of a core complex and an interacting complex [10,11]. The core complex consists of three subunits, methyltransferase like 3 and 14 (METTL3/14), and Wilms tumor 1-associated protein (WTAP) [11,12,13]. METTL3 is the only catalytic component, METTL14 is an adaptor required for METTL3 activity and RNA binding, and WTAP is a regulator that interacts with METTL3 to regulate its localization and activity [14]. Components of the interacting complex include Vir-like m^6^A methyltransferase associated (VIRMA), Zinc Finger CCCH-Type Containing 13 (ZC3H13), RNA-binding motif protein 15 (RBM15), and Casitas B-lineage proto-oncogene like 1 (CBLL1), which anchor the core complex of m^6^A methyltransferase to nuclear speckles and exert distinct regulatory functions [10,11,15,16,17].

VIRMA was first isolated in *Drosophila melanogaster,* named as VIRILIZER (Vir), which performs a function in sex determination and participates in mRNA alternative splicing [18,19,20,21]. Mammalian KIAA1429, a homologue of *D. melanogaster* VIRILIZER (Vir), interacts with m^6^A methyltransferase METTL3 and WTAP, is a member of mRNA m^6^A writer proteins, and is required for m^6^A writer activity [15,20,22,23]. In addition, VIRMA can promote the progression of cancer and is associated with poor survival in multiple types of cancer [24,25,26]. *Arabidopsis* VIR, which is also named VIRILIZER/KIAA1429/EMB2016, is characterized as a splicing/methylation factor, and is a member of the mRNA m^6^A methyltransferase complex [27]. Knock-down of VIR affects plant architecture and root vascular development and displays salt-hypersensitive phenotypes in an m^6^A-dependent manner [27,28]. Although AtVIR plays important roles during plant growth and development, its other biological functions in plants have yet to be determined.

Leaves are the primary organs that provide energy for all organs through sugar production during photosynthesis, and as such, have a pivotal role in plant growth and development [29]. Leaves are initiated at the shoot apical meristems (SAM), where later leaf polarity is established along three-dimensional axes [30]. After the leaf polarity is established, leaves grow through cell proliferation and cell expansion to acquire their final size and shape in most dicotyledonous species [29,30]. Many genes regulate size, shape, and differentiation during leaf development [29,30]. In *Arabidopsis*, SHOOTMERISTEMLESS (STM) and WUSCHEL (WUS) are the key regulators to maintain SAM, and determine the leaf initiation [29,31]. CUP-SHAPED COTYLEDON1/2/3 (CUC1/2/3), LATERAL ORGAN BOUNDARIES (LOB), and KNOTTED-LIKE FROM ARABIDOPSIS THALIANA1 (KNAT1) transcription factor are the major transcription factors required to initiate SAM and boundary formation. In the leaf polarity establishment and leaf outgrowth stage, ASYMMETRIC LEAVES1/2 (AS1/2), PRESSED FLOWER (PRS), auxin response factor 2/3/4 (ARF2/3/4), WUSCHEL-RELATED HOMEOBOX1 (WOX1), and YABBY (YAB) are the major regulators to maintain adaxial–abaxial polarity and establish proximal–distal polarity [29]. In the leaf flattening and expansion (cell proliferation and cell expansion) stage, the growth-regulating factor (GRF)-GRF interaction factor (GIF) module, the PRS/WOX1/3 module, and the miR319-Class II TCP transcription factors (TCP2/3/4/10/24)-NGATHA (NGA) module control the switch from cell proliferation to cell differentiation and then determine the leaf expansion and differentiation. In addition to transcription factors, mechanical forces are also important for leaf morphogenesis [32,33]. Zhao et al. (2020) showed that microtubules and cellulose microfibrils align along the main stress direction of internal walls to mediate anisotropic growth. Microtubule-mediated mechanical feedback amplifies an initial asymmetry and maintains directional growth [34].

Leaves perform photosynthesis and produce their own energy and carbon sources for plant growth and development. Chloroplasts are the central organelles performing photosynthesis and producing sugar. Hudik et al. (2014) demonstrated that impaired chloroplast differentiation affects leaf cell proliferation and induces an early onset of cell differentiation [35]. Chloroplasts produce sugar signals, which trigger the transition to cell expansion during leaf development [36]. As seen in the maize leaf, chloroplast biogenic processes clearly coincide with leaf development [37,38]. Moreover, they also coincide with the expression of transcription factors that are linked to light responses and plastid-to-nucleus communication [37,38]. GOLDEN2-LIKE 1 and 2 (GLK1/2), GATA NITRATE-INDUCIBLE CARBONMETABOLISM-INVOLVED (GNC), and CYTOKININRESPONSIVE GATA FACTOR 1 (CGA1) are key chloroplast biogenesis factors that regulate the transcription of chloroplast—associated nuclear genes (CpANGs) and photosynthesis-associated nuclear genes (PhANGs) [39]. In addition, plastid transcription is carried out by both nuclear-encoded polymerases (NEPs) and plastid-encoded polymerases (PEPs) [40]. A well characterized group of plastid SIGMA (SIGs) factors and plastid transcriptionally active factors (pTACs) control the initiation of PEP-mediated transcription of chloroplast genes [41,42]. These transcription factors form a complex network regulating the transcription of CpANGs and PhANGs to control chloroplast biogenesis and regulate leaf development [39].

In this study, we characterized *GhVIRs* in cotton and analyzed gene expression patterns in various developing cotton tissues; then, we further analyzed the function of *GhVIRs* in leaf development. Our results demonstrate that GhVIRs are m^6^A regulatory factors that regulate leaf development through integrating multiple signaling pathways, providing a valuable reference for cotton leaf development and photosynthetic efficiency improvement.

## 2. Results

### 2.1. Identification and Characterization of G. hirsutum VIR Genes

Plant VIRs were first identified in *Arabidopsis* [27]. A BLASTP search was performed using the protein sequence of *Arabidopsis* VIRMA as query sequences on the website of COTTONGEN; two *G. hirsutum* VIR sequences were identified, one comes from the cotton A subgenome Gh_A04G1327, named as *GhVIR-A*, and another one comes from the cotton D subgenome Gh_D04G0886, named as *GhVIR-D*. The two *GhVIRs* share 98.5% nucleotide sequence identity in their open reading frame region. The basic information of these two GhVIRs is shown in Table 1.

We obtained VIR homologs in different land plants and algae from Phytozome 13, including *Gossypium* plants (*Gossypium barbadense* v1.1, *Gossypium darwinii* v1.1, *Gossypium hirsutum* v2.1, *Gossypium hirsutum* CSX8308 v1.1, *Gossypium hirsutum* UA48 v1.1, *Gossypium hirsutum* UGA230 v1.1, *Gossypium mustelinum* v1.1, *Gossypium raimondii* v2.1, and *Gossypium tomentosum* v1.1), dicotyledons (*Arabidopsis* V11, *Aquilegia coerulea* v3.1, *Amaranthus hypochondriacus* v2.1, *Spinacia oleracea* Spov3, *Solanum lycopersicum* ITAG4.0, *Vitis vinifera* v2.1, *Glycine max* Wm82.a4.v1, *Phaseolus vulgaris* v2.1, *Vigna unguiculata*, *Populus trichocarpa* v4.1, *Theobroma cacao* v2.1, and *Brassica rapa* FPsc v1.3), monocotyledons (*Ananas comosus* v3, *Oryza sativa* v7.0, *Brachypodium distachyon* v3.1, *Sorghum bicolor* v3.1.1, and *Zea mays* RefGen_V4), moss (*Physcomitrella patens* v3.3), fern (*Selaginella moellendorffii* v1.0), and algae (*Chlamydomonas reinhardtii* v5.6 and *Micromonas pusilla* CCMP1545 v3.0) (Appendix A). We found that there is no VIR gene in algae and fern genomes. Except for *Glycine max*, most diploid plants contain one VIR in their genome. Apart from *Gossypium barbadense*, which contains three VIR members, other chosen tetraploid plants contain two VIR genes in their genome (Figure 1, Appendix A). To understand the evolutionary relationships among VIRs in plants, VIR homologs in different species were analyzed in detail using neighbor-joining methods, and the unrooted phylogenetic tree was constructed (Figure 1A). The VIR homologs could be classified into two clades, namely clade I (monocot and moss) and clade II (eudicot). Plant VIR proteins contain four conserved domains: an N-terminal region with approximately 215 AA (1–215 AA), a middle domain I (564–1291 AA), a middle domain II in the 1494–1764 AA and a C-terminal region with approximately 1878–2138 AA of AtVIR (Appendix A). We further checked the conversed motifs by the MEME suite (https://meme-suite.org/meme/tools/meme, accessed on 30 July 2022) and AA identities by DNAStar software. The results also show plant VIRs share similar motifs and 38~98% identity between each other (Appendix A).

### 2.2. Expression Analysis of GhVIRs in Upland Cotton

*GhVIR-A* and *GhVIR-D* share similar expression patterns in transcriptome data, which are present in all checked tissues/organs and are relatively higher expressed in main stem apex (MSA), pistil, stem, leaf, and 3 DPA (day post anthesis) ovule, but relatively weak expression of these genes are detected in the other tissues of cotton (Figure 2A). This tissue/organ-ubiquitous expression pattern indicates that GhVIR-A/D may function in different cotton tissues, especially in the cotton main stem apex and leaf.

We further carried out quantitative real-time PCR (qRT-PCR) to investigate the expression profiles of *GhVIR-A/D* (Figure 2B). The results revealed that *GhVIR* genes were expressed in all organs and tissues detected. *GhVIRs* were expressed relatively higher in MSA, leaf, and 12 DPA fiber (Figure 2B).

Furthermore, we checked the expression of *GhVIRs* in different species with different leaf shapes (WT, okra, and super okra). The qRT-PCR results show *GhVIRs* share a higher expression level in leaf of okra and super okra species than Coker 312 (Figure 2C). We also examined the expression of *GhVIRs* in the MSA of the selected allopolyploid cottons with different plant heights and found that the expression levels of *GhVIRs* in MSA were correlated with plant height (Figure 2D,E). These results indicate that *GhVIRs* may play roles in leaf and MSA development.

### 2.3. GhVIR Function in Regulating Cotton Architecture and Leaf Development

In *Arabidopsis*, *At**VIR* has been reported to play an essential role in developmental decisions during pattern formation and is required for embryo development and salt stress tolerance [27,28,43]. GhVIRs shared 59.2% identity with AtVIR (Appendix A). In our studies, *GhVIR* gene transcripts were highly detected in the leaves and MSA and co-related with different plant height and leaf shape (Figure 2C–E). Therefore, we speculate that *GhVIRs* are potential candidates to regulate pattern formation of cotton tissues, especially for cotton MSA and leaf morphogenesis.

Virus-based expression tools are good strategies that allow for functional gene analyses in cotton. Two virus-based systems are reported: the disarmed geminivirus cotton leaf crumple virus (dCLCrV) and tobravirus tobacco rattle virus (TRV) are used for virus-induced gene silencing (VIGS) in cotton [44,45,46,47,48]. Furthermore, McGarry et al. (2016) compared the strength and duration of dCLCRV and TRV systems in silencing identical gene sequences. They found systemic silencing by TRV systems in cotton has higher potency but shorter duration than dCLCRV systems [48]. Therefore, we chose the two VIGS methods to analyze the functions of GhVIR during cotton growth and development.

Firstly, we specifically silenced the gene expression of *GhVIRs* by a TRV-based VIGS technique; the indicator plant (*TRV:CLA1*) exhibited an albino phenotype in their young leaves (Appendix A). Then, we used the qRT-PCR to confirm the effect of the silenced genes. The results indicate that the expression of *GhVIRs* was remarkably reduced in cotton leaves of *TRV:**GhVIRs* plants compared to negative control plants (*TRV:00* plants) (Figure 3A). We also checked mRNA m^6^A levels by dot blotting and colorimetric methods in *GhVIR-VIGS* and control plants (Figure 3B). The results show that mRNA m^6^A levels are significantly lower in *GhVIR-VIGS* leaves compared with *TRV:00* plants. For phenotype analysis, we found that *GhVIR-VIGS* lines exhibited a very dwarfed phenotype with smaller, down-curly, wrinkled, and browning leaves compared to those of control plants (*TRV:00* plants) (Figure 3C,D). As the plants developed, the second leaf of the *TRV:GhVIR* plants would drop off from the plants when the control plants reached the four- or five-leaf stage (Appendix A). These plants would die when the control plants reached the six-leaf stage. Then, we used the dCLCrV system to analyze the function of *GhVIRs,* as the method can silence target genes in the whole life cycle of cotton plants. As shown in Figure 4A, *dCLCr**V**:PDS* plants are the indicator plants which exhibited albino phenotypes in their young leaves. The qRT-PCR results also show that expression of *GhVIRs* was silenced (Figure 4B). Similar to the silencing results of the TRV system, *dCLCrVA:GhVIR* plants also showed a smaller, wrinkled, and down-curly phenotype (Figure 4H). We noted that knockdown of *GhVIRs* resulted in abnormal development of terminal buds, many leaflets clustered at the tip and branches, and the plants showed an overall dwarfed and multi-branched phenotype; these plants’ growth and development were strongly inhibited (Figure 4C–G). We found all of *dCLCr**V-GhVIRs* plants are incapable of transitioning from vegetative to reproductive stage (Figure 4I), some of the *dCLCr**V-GhVIRs* plants drop off their leaves and die, and the same for *TRV:GhVIR* plants. These results substantiate that *GhVIRs* are essential for main stem apex and leaf morphogenesis and determine the plant architecture.

Shao et al. (2021) reviewed m^6^A modification in plants and proposed that the regulatory mechanism of m^6^A underlying leaf development remains to be elucidated [7]. For this reason, we mainly focused on the effect of *GhVIR* on leaf development. Paraffin section results showed that leaf thickness of *TRV:**GhVIR* and *d**CLCr**V-GhVIRs* plants are thinner, and leaf cells structurally abnormal, irregular, and disordered, and fewer vessel cells in leaf veins appeared compared with control plants (Figure 4J and Figure 5A–D). Next, we checked the leaf cell shape by super depth of field microscope, and found the pavement cells of *TRV:GhVIR* plants are smaller and abnormal compared with *TRV:00* plants (Figure 5E,F). Finally, we counted the leaf area, leaf epidermal cell size, and total cell number of the whole leaf in *TRV:GhVIR* plants and *TRV:00* plants. We found the leaf cell size, total cell number, and the leaf area of *TRV:GhVIR* plants are about one third, half, and one sixth compared with *TRV:00* plants, respectively (Figure 5G–I). These results suggest that GhVIRs regulate leaf cell shade, proliferation, and expansion and then influence leaf morphogenesis.

### 2.4. GhVIR Regulates Expression of Genes Related to Leaf Development, Chloroplast Biosynthesis and Photosynthesis

To uncover the differentially expressed genes (DEGs) between *GhVIR* VIGS lines and control lines (*TRV:00* lines), we performed transcriptomic analysis to find genes regulated by GhVIRs in cotton leaves. A total of 106,987,495 and 109,836,500 paired-end 150-bp reads were generated from three biological repeat libraries of leaf of *GhVIRMA* VIGS lines and control lines (*TRV:00* lines), respectively. Approximately 91.4% of the reads mapped to the upland cotton genome. In total, 7437 DEGs were identified (Figure 6A). Among the 7437 DEGs, 4047 genes were up-regulated and 3390 were down-regulated in the *GhVIR* VIGS lines (Figure 6A, Appendix A). KEGG pathway analysis showed that the genes up-regulated in *GhVIR* VIGS lines were mainly enriched in plant hormone signal transduction, biosynthesis of amino acids, plant-pathogen interaction, carbon metabolism, phenylpropanoid biosynthesis, glutathione metabolism, flavonoid biosynthesis, amino sugar and nucleotide sugar metabolism, glycolysis/gluconeogenesis, and cysteine and methionine metabolism (Figure 6B). By contrast, down-regulated genes were involved in photosynthesis—antenna proteins, photosynthesis, ribosome, carbon metabolism, plant hormone signal transduction, starch and sucrose metabolism, biosynthesis of amino acids, carbon fixation in photosynthetic organisms, glutathione metabolism, alpha-Linolenic acid metabolism, glycine, serine and threonine metabolism, porphyrin and chlorophyll metabolism, the pentose phosphate pathway, fructose and mannose metabolism, carotenoid biosynthesis, and fatty acid elongation (Figure 6C).

Further analysis of these DEGs revealed that they included not only chloroplast development and transcription factors (TFs), but a number of genes related to cytoskeletal organization, cell wall biosynthesis, plant hormone metabolism and catabolism, and so on (Appendix A). Among the up-regulated genes, we found 36 WRKY TFs (such as homologs of *WRKY75*), 22 NAC TFs (homologs of *ANAC072*), 20 ERF TFs (such as *ERF1B*), eight AT-hook TFs, and three *C2H2*-like (*ZAT-like*) TFs; the functions of these TFs mainly relate to cell defense, abiotic or biotic stress, and immunity. The genes differentially expressed also include ethylene biosynthesis genes, signaling pathway genes and receptors (*ACOs*, *ACSs*, *EIN2/4*, *EBFs*, *ETR1*, and *ETR2*), gibberellin (GA) catabolic enzymes and GA receptors (*GA2ox8*, *GA3ox1*, *GID1Bs*, *and GID1Cs*), and five topless-related (homologs of *AtTPL*) genes that may be vital for hormone metabolism and signaling pathways. On the other hand, the down-regulated genes encode CIN-group TCPs, YABBYs, and squamosa promoter-binding-like protein (SPLs) TFs, transcription termination factors (mTERFs) for chloroplast genes, and ten chloroplast RNA polymerase sigma factors (Sigma-70), and include hundreds of chloroplast genes (these genes encode 30S/50S ribosomal proteins, chlorophyll a-b binding proteins, ferredoxin and ferredoxin--NADP reductases, photosystem I/II related proteins, thylakoid proteins, PsbPs, PsbDs, and PsbAs), and dozens of cell wall and skeleton related genes (these genes encode pectinesterases, lipid transfer proteins, xyloglucan endo-transglycosylases (XETs), profilins, actins, actin-depolymerizing factors, alpha-tubulins, and beta-tubulins) (Appendix A). We further verified the expression levels of these genes in the leaves of *TRV:GhVIR* and *TRV:00* plants. As shown in Figure 6C, qRT-PCR results were consistent with the RNA-Seq analysis. These data indicate that GhVIRs may directly or indirectly regulate a number of downstream genes that have an effect on leaf development, chloroplast biosynthesis, photosynthesis, cell defense, and cell wall and cell skeleton biosynthesis.

We also observed the expression of some other regulators in chloroplast development in the leaves of *TRV:00* and *TRV:GhVIR* plants (Appendix A). Chloroplast development regulators *GhGLKs-1/-2* (*GhGLK1/2*), *GhGCN 1* (*GhGCN1*), were slightly upregulated in *TRV:GhVIRs* plants [39] (Appendix A). These results demonstrate that GhVIRs regulate leaf development independent of *GhGLK1/2* and *GhGCN1/2* pathways.

### 2.5. GhVIR Function in Chloroplast Biosynthesis and Photosynthesis

Transcriptome data analysis showed that the expression of genes related to chloroplast biosynthesis, photosystem I/II, and the electron transport chain were significantly down-regulated, suggesting that *GhVIRs* may regulate chloroplast development, thus affecting leaf photosynthesis. Therefore, we examined the structure of chloroplasts by transmission electron microscopy. The results show that chloroplasts in *TRV:GhVIR* plants were severely damaged and distorted, and thylakoid of these plants appeared to be damaged or ruptured, and even degraded compared with *TRV:00* plants (Figure 7A–C). We further examined the chloroplast pigment contents in leaves of *TRV:00* and *TRV:GhVIR* plants. The results show the contents of chlorophyll a, chlorophyll b, and carotenoid in the leaves of *TRV:GhVIR* plants with the same weight were much higher than those of *TRV:00* plants (Figure 7D). At the same time, we also measured the activities of photosystem II in the leaves of *TRV:00* and *TRV:GhVIR* plants by using chlorophyll a fluorescence (OJIP) transients (Handy PEA Hansatech Instruments Ltd., King’s Lynn, UK). OJIP transients are defined by the O, J, I and P steps, which correspond to the redox state of photosystem II [49,50]. The experiments on the cotton leaves were performed at two time points: one was 10 days after injection of the *Agrobacterium* solution (TRV system); another time point was 13 days after injection. As shown in Figure 7E, there was no significant difference in OJIP curves between leaves of *TRV:00* and *TRV:GhVIR* plants at 10 days post-injection when there was no browning on the leaf surface. However, after 13 days of injection, browning appeared on the leaf surface of *TRV:GhVIR* plants (Figure 3C,D, arrowheads), so we measured chlorophyll fluorescence values of plant leaves again. We found that the chlorophyll fluorescence values of leaves of *TRV:GhVIR* plants were reduced at the I, P point compared to *TRV:00* plants (Figure 7F). The results show that the activity of photosystem II was significantly reduced after silencing *GhVIR* genes compared with control plants (Figure 7F). This result is consistent with the transcriptome finding that a large number of differential genes associated with chloroplast biosynthesis and the photosynthetic system are significantly down-regulated in *TRV:GhVIR* plants. These results indicate that GhVIR regulates chloroplast development, but the target genes and regulatory mechanisms of m^6^A modification need to be further explored.

## 3. Discussion

RNA modifications constitute an essential layer of gene regulation in living organisms. As the most prevalent internal modification of eukaryotic mRNAs, N6-methyladenosine (m^6^A) exists in many plant species and requires the evolutionarily conserved methyltransferases for writing m^6^A. In *Arabidopsis*, mRNA m^6^A modification is mainly mediated by a methyltransferase complex composed of MTA (the ortholog of mammalian METTL3), MTB (the ortholog of mammalian METTL14), FIP37 (the ortholog of mammalian WTAP), VIRMA, and an E3 ubiquitin ligase HAKAI [3,27]. *Arabidopsis* VIRMA has been shown to play important roles in developmental decisions during pattern formation and salt stress tolerance, but its mechanism in plant growth and development is still unclear [27,28].

In this study, we identified two *VIRMA* genes from the upland cotton genome and analyzed their evolutionary relationships with homologs of other plants (Figure 1A). Based on the phylogenetic analysis, plant VIRMA proteins are grouped into two clades, monocot and eudicot (Figure 1A). There is no *VIR* gene in algae genomes, but it appears in moss genomes, suggesting that VIR may have arisen from the evolution of vascular plants from marine to terrestrial. In addition, except *Glycine max,* most diploid plants contain one *VIR* gene in their genome, and tetraploid plants contain two *VIR* copies. The results suggest that plant *VIR* gene duplication mainly depends on the duplication of genome doubling. Different VIRs share high homology (38–98% identity each VIRs) with four conserved domains (Appendix A), indicating that plant VIRs may have conserved biological function, but their functions in plants need to be further explored. Plant VIRs also share four relatively conversed domains with *Drosophila melanogaster* VIRMA (DmVIR) and *Homosapiens* VIRMA (HsVIR), and have a specific C-terminal domain (the fourth domain in plant VIRs) that is not present in DmVIR and HsVIR (Appendix A). These conversed domains may share a similar function in plant and animal VIRs, and the C-terminal domain in plant VIRs may play special roles in plant development. Furthermore, *GhVIRs* are ubiquitously expressed in different tissues of cotton and relative highly expressed in cotton leaf and MSA (Figure 2). In this study, knocking down *GhVIR* expression, the *GhVIR*-silencing plants exhibited extremely short and abnormal leaf morphogenesis phenotypes (Figure 3 and Figure 4). The phenotype of *GhVIR*-silencing plants is similar to *vir-1* mutant and AtVIR RNAi *Arabidopsis* plants [27]. The results demonstrate that GhVIRs share a conservative expression pattern and function with AtVIR.

Leaves are the primary organs for photosynthesis and therefore have a pivotal role in plant growth and development. Leaf development is a multifactorial and dynamic process involving leaf initiation, leaf polarity determination, and leaf flattening and expansion (leaf blade initiation and intercalary growth) [29]. Studies have reported that m^6^A plays an important role in *Arabidopsis* leaf development [51,52]; however, its molecular mechanism needs to be further elucidated [7]. As a component of the m^6^A methyltransferase complex, we speculated that VIR proteins may play important roles in leaf morphogenesis. We chose the VIGS method to silence the expression of *GhVIRs* and then elucidate the functions of *GhVIRs* in cotton leaf development. Indeed, silencing *GhVIRs* by VIGS led to a smaller, wrinkled, and down-curly leaves (Figure 3 and Figure 4). Microscopy results show that leaves of *TRV:GhVIR* plants share smaller cell size, fewer cell numbers, and less leaf area compared to *TRV:00* plants (Figure 5). These results indicate that *GhVIRs* may regulate leaf polarity determination, cell proliferation and expansion, and then influence the leaf morphogenesis (including leaf shape, flattening and expansion). Additionally, expression of some leaf polarity determination, and cell proliferation and expansion related genes are down-regulated in *GhVIRs*-silencing cotton, which may contribute to produce smaller, wrinkled, and down-curly leaves (Figure 6). For example, expression of some cytoskeleton and cell wall related genes, including cotton alpha-tubulin6 (GhaTUB6) and beta-tubulin6 (GhbTUB6), were significantly down-regulated in *GhVIRs*-silencing leaves (Figure 6, Appendix A). Mechanical forces are also important for leaf morphogenesis [32,33,34]. Zhao et al. found microtubule-mediated mechanical feedback amplifies an initial asymmetry and maintains leaf directional growth and contributes to leaf flattening [34]. The leaf morphogenesis depends on cortical microtubule mediated cellulose deposition along the main predicted stress orientations, in particular, along the adaxial-abaxial axis in internal cell walls [34]. Therefore, GhVIR-mediated downregulation of some leaf cytoskeleton and cell wall-related genes may be a cause of abnormal leaf morphogenesis. We also found some homologs of key regulators in determining *Arabidopsis* leaf polarity establishment and leaf flattening and expansion, including axial regulator YABBY, auxin response factors (ARFs), growth-regulating factors (GRFs), GRF interacting factors (GIFs), and CIN-class TCP transcription factors (TCP3/4/10), in *GhVIR*-silencing leaves were down-regulated (Figure 6, Appendix A). In *Arabidopsis*, these regulators control the switch from cell proliferation to cell differentiation and then determine the leaf expansion and differentiation [29]. Downregulation of genes involved in leaf cell proliferation and differentiation leads to significant change in cell size, cell number, and cell shape, thereby determining cotton leaf morphogenesis in *GhVIR*-silencing plants. Plant hormones also play important roles in leaf development [31]. In our experiments, we also found about one hundred differential expression genes involved in plant hormone biosynthesis/catabolism and signaling pathways (Appendix A). The results suggest that GhVIRs regulate leaf morphogenesis by plant hormones mediating signaling pathways [31].

Chloroplasts perform photosynthesis and produce energy and carbon sources, which play important roles in plant leaf development. Chloroplasts produce sugar signals, which trigger the transition to cell expansion during leaf development [36]. Chloroplast differentiation is associated with leaf cell proliferation and cell differentiation [37]. With impaired or lost chloroplasts, leaves become albino, just like the leaves of *TRV:GhCLA1* or *dCLCrV:GhPDS* plants (Figure 4A and Appendix A). In our experiments, we found that leaves of *GhVIR*-silencing plants browned after injection with VIGS *Agrobacterium* to cotton cotyledon over thirteen days (Figure 3). We speculate that silencing *GhVIRs* may influence chloroplast structure and/or photosynthesis. Indeed, transmission electron microscope results show that the chloroplast structure of *TRV:GhVIR* plants was damaged, and the thylakoid structure was broken and even degraded (Figure 7A–C). *TRV:GhVIR* plant chloroplast pigment content is increased in leaves with the same weight compared to *TRV:00* plants (Figure 7D). Furthermore, the photosynthetic efficiency of photosystem II in *TRV:GhVIR* plants was reduced when the leaves of *TRV:GhVIR* plants were browning (Figure 7F). At the transcription level, we found hundreds of genes involved in chloroplast biosynthesis and the photosynthesis system, chloroplast ribosome biosynthesis (some genes encode 30S and 50S ribosome proteins), transcription regulators of chloroplast genes (RNA polymerase sigma factor sigA, plastid transcriptionally active 16 and transcription termination factor MTERF1) were significantly down-regulated in *TRV:GhVIR* plants (Figure 6, Appendix A). Therefore, we speculate that *GhVIRs* regulate chloroplast structure and photosynthetic efficiency by influencing plastid transcription and regulating the expression of chloroplast- and photosynthesis—associated nuclear genes (*CpANGs* and *PhANGs*), which determine leaf morphogenesis and even browning and dying. Plastid transcription is carried out by both nuclear-encoded polymerases (NEPs) and plastid-encoded polymerases (PEPs) [40]. Transcription factors such as GOLDEN2-LIKE 1 and 2 (GLK1&2), GATA NITRATE-INDUCIBLE CARBONMETABOLISM-INVOLVED (GNC) and CYTOKININRESPONSIVE GATA FACTOR 1 (CGA1) are key factors involved in chloroplast biogenesis, which regulate the transcription of *CpANGs* and *PhANGs* [39]. In addition, plastid SIGMA (SIGs) factors and plastid transcriptionally active factors (pTACs) control initiation of PEP-mediated transcription of chloroplast genes [41,42]. In our study, we found the expression of *GhGLK1/2* and *GhGNC/CGA1* were slightly higher in *TRV:GhVIR* plants than the control plants (Appendix A) [39]. *GhSIGs* and *GhpTACs* were significantly reduced in *TRV:GhVIR* plants. We speculate that GhVIR regulate chloroplast biogenesis and plastid transcription independent of *GhGLK1/2* and *GhGNC/CGA1* pathways mediating to regulate the expression of *CpANGs* and *PhANGs*. However, the mechanism for GhVIR regulation of chloroplast-dependent leaf development needs to be further explored.

In summary, we identified and characterized cotton VIR genes and illustrated their expression patterns. We found that GhVIRs act as a component of the m^6^A methyltransferase complex, regulating the mRNA m^6^A modification level in cotton. GhVIRs regulate leaf morphogenesis and affect plant height and branching. GhVIRs may act as a key post-transcriptional switch to regulate a cascade of transcription factors, some cytoskeleton and cell wall related genes, genes involved in plant hormone biosynthesis/catabolism and signaling pathways, genes involved in chloroplast biosynthesis and the photosynthesis system, thereby regulating leaf morphogenesis by controlling chloroplast-dependent and independent pathways. This study provides important clues for studying the function of plant VIRs, deepens our understanding of cotton leaf regulation, and establishes a resource for cotton breeding.

## 4. Materials and Methods

### 4.1. Plant Materials

The seeds of upland cotton (*G. hirsutum*) were surface sterilized with 70% (*v/v*) ethanol for 1 min, and then with 10% hydrogen peroxide for 2 h, followed by washing with sterile water several times. The sterilized seeds were germinated in one-half strength Murashige and Skoog (MS) medium (16-h-light/8-h-dark cycle, 28 °C), and seedlings were transplanted to the soil for further growth. The roots, stems, main stem apex (MSA, apex length is about 5 mm), young leaves of three-leaf stage cotton plants, and different developmental stage cotton fibers and 10 DPA (days post-anthesis) ovules after flowering were harvested for RNA extraction.

### 4.2. Sequence Analysis

A BLASTP search was performed using the protein sequence of *Arabidopsis* VIRMA [27] as query sequences on the website of COTTONGEN (https://www.cottongen.org/blast/protein/protein, accessed on 30 July 2022) by choosing the *G. hirsutum* Novogene Bioinformatics Technology (NBI) protein database, and 2 GhVIRMA sequences were identified. The chromosomal location, amino acid length, protein molecular mass, and isoelectric point of the 2 GhVIRMAs were analyzed using COTTONGEN (https://www.cottongen.org/find/genes, accessed on 30 July 2022) and ExPASy ProtParam (http://us.expasy.org/tools/protparam.html, accessed on 30 July 2022). DNA and protein sequences were analyzed using DNASTAR software (DNAStar V7.1, Madison, WI, USA). The MEME Server Program (https://meme-suite.org/meme/, accessed on 30 July 2022) was used to analyze the conversed motifs in GhVIRMA, OsVIRMA, and AtVIRMA amino acid sequences.

### 4.3. Phylogenetic Analysis

The protein sequence of *Arabidopsis* VIRMA [27] was used as queries to identify different cotton species and different plant species from Phytozome v13 (https://phytozome-next.jgi.doe.gov/pz/portal.html, accessed on 30 July 2022). A phylogenetic tree of deduced plant VIRMA amino acid sequences was constructed using the Neighbor-Joining algorithm with default parameters, with 1000 bootstrap replicates in MEGA_X_10.2.4 (https://www.megasoftware.net, accessed on 25 August 2021, MEME version 5.4.1) [53].

### 4.4. Heat-Map Analysis of Gene Expression

The expression patterns of *GhVIR-A/D* were investigated by analyzing gene expression of different cotton tissues, including roots, stems, leaves, main stem apex, torus, calycles, pistils, petals, stamens, ovules, and fibers at several developmental stages using transcriptome data [45,46]. The RPKM (reads per kb per million reads) values denoting the expression levels of *GhVIRMA* genes were isolated from a comprehensive profile of the TM-1 transcriptome data (http://structuralbiology.cau.edu.cn/gossypium, accessed on 30 July 2022) [54,55], and the expression data of MSA were generated in this study. A heat map analysis was performed using TBtools (https://github.com/CJ-Chen/TBtools, accessed on 10 July 2022, TBtools_windows-x64 Version No.1.098761) [56].

### 4.5. Expression Analysis

Total RNA was extracted from cotton roots, stems, leaves, ovules, and the different developmental fibers after flowering using the RNAprep Pure Plant kit (TIANGEN, Beijing, China) and reverse transcribed using Moloney murine leukemia virus reverse transcriptase (Promega, Madison, WI, USA) according to the manufacturer’s instructions. QRT-PCR was performed using the MJ Research DNA Engine Option 2 detection system with the fluorescent intercalating dye SYBR-Green (Toyobo, Osaka, Japan). The relative expression levels were normalized to a cotton polyubiquitin gene (*GhUBI1*, GenBank accession no. EU604080).

A two-step PCR procedure was performed in all experiments using a method described earlier [57]. The relative target gene expression was determined using the comparative cycle threshold method. To achieve optimal amplification, PCR conditions for each set of primers were optimized for annealing temperature and Mg^2+^ concentration. Data presented in the qRT-PCR analysis are the mean and standard deviation of three biological replicates of plant materials and three technical replicates in each biological sample using gene-specific primers. Primers used are listed in Appendix A.

### 4.6. Agrobacterium Tumefaciens-Mediated VIGS

Binary vectors were introduced to the *Agrobacterium tumefasciens* strain GV3101 by electroporation. Single colonies were used to inoculate cultures. TRV and dCLCrV are bipartite viruses, and equal volumes of inoculum harboring each component of each virus were mixed and infiltrated into two fully expanded cotyledons of 10-day-old cotton plants grown at 22–24 °C using a 1 mL syringe as described previously [46,48]. At least fifteen plants were inoculated for each construct. Inoculated seedlings were covered overnight at 25 °C, and maintained in a glasshouse at 25 °C under 16 h light and 8 h dark for 10 days (2-leaf stage) to test the silencing efficiency. The *TRV2:GhCLA1* and *dCLCrV:GhPDS* construct was included as a visual marker for VIGS efficiency. Primers used are listed in Appendix A.

### 4.7. RNA-Seq Analysis

Total RNA was extracted from the third leaf of three independent *GhVIR* VIGS lines and three TRV2 control cotton plants, and 3μg RNA per sample was used for constructing cotton RNA-seq libraries. Sequencing libraries were generated using NEBNext^®^ Ultra™ RNA Library Prep Kit for Illumina^®^ (NEB, Ipswich, MA, USA) and sequenced on an Illumina Hiseq platform; 125 bp/150 bp paired-end reads were generated (Biomarker Technologies Technology Co., Ltd., Beijing, China, https://www.bmkgene.com, accessed on 21 August 2018). Three biological repeats were performed for the RNA-Seq experiment. Then, the trinity program was used to assemble high quality reads for each sample. Functional annotations for the assembled unigenes were performed by BLAST similarity search against *Gossypium hirsutum* L. acc. TM-1 (AD1) genome NAU-NBI (Nanjing Agricultural University-Novogene Bioinformatics Technology) assembly v1.1 and annotation v1.1 (https://www.cottongen.org, accessed on 1 May 2022) [55], NCBI, PANTHER, GO, KEGG or domain search against Pfam.

For analysis of differentially expressed genes (DEGs), HTSeq v0.6.1 was used to count the read numbers mapped to each gene. Then, the FPKM of each gene was calculated based on the length of the gene and read count mapped to the gene. With the read count of unigenes, DESeq R package (1.18.0) was performed to generate statistical information of DEGs, such as expression level, fold change, *p*-value, and FDR. The DEGs were defined by following conditions: fold change greater than 2; *p*-value and FDR lower than 0.01. To analyze DEGs between the *GhVIRMA* VIGS lines and TRV2 control lines, the double best hits were identified and the expression was compared between double best hit pairs.

### 4.8. Measurement of m^6^A Content

To estimate the relative m^6^A content of RNA, EpiQuik m^6^A RNA Methylation Quantification Kit (Epigentek, New York, NY, USA) was used. Briefly, equal volumes of RNA solution (4–8 mL) were processed according to the manufacturer’s instructions, aside with the negative, positive, and standard controls.

### 4.9. Dot Blot Analysis of m^6^A Level

Purified mRNA was first denatured by heating at 65 °C for 2 min, followed by chilling on ice directly. mRNA was spotted on a Hybond-N+ membrane (Amersham, USA) optimized for nucleic acid transfer. After UV crosslinking in a Stratalinker 2400 UV Crosslinker (Stratagene, La Jolla, CA, USA), the membrane was washed by PBST buffer (PBS with Tween-20), blocked with 5% of non-fat milk in PBST, and incubated with m^6^A-specific antibody (1:250; No. 202003, Synaptic Systems, Goettingen, Germany) overnight at 4 °C. After incubating with horseradish-peroxidase-conjugated anti-rabbit IgG secondary antibody (Santa Cruz, CA, USA), the membrane was visualized by an ECL Western Blotting Detection Kit (Thermo Fisher, Waltham, MA, USA).

### 4.10. Microscopy

At the 2-leaf stage, leaves of TRV:00/TRV:GhVIR plants were analyzed by transmission electron microscopy according to a method described by Zhang et al. (2018) [58]. Treated leaf samples (cross sections) were examined and photographed under a transmission electron microscope (Hitachi, Tokyo, Japan). Additionally, paraffin-embedded cross-sections of leaves were observed and photographed under a super depth of field microscope as described previously with a little modification [59].

### 4.11. Determination of Pigment Content in Chloroplast

An amount of 0.2 g of fresh leaves was taken and ground; then, 1 mL of pre-cooled methanol was added and mixed thoroughly. The supernatant was placed at −20 °C for 1–2 h, taken out, and centrifuged at 4 °C and 12,000 rpm for 2 min. The volume of the supernatant was recorded; an appropriate amount of the supernatant was taken and put in a multi-function microplate reader, and its absorbance values were measured at 665 nm, 649 nm, and 470 nm, respectively. According to Lambert Beer’s law, the content of chlorophyll a, b and carotenoids was calculated. The determination result is based on three biological repetitions.

### 4.12. Determination of Photosynthetic Efficiency of Photosystem II by Handy PEA

OJIP transients in cotton leaves were measured when the second leaf of TRV:GhVIR was fully expanded. The first measurement was carried out 10 days after the injection of the Agrobacterium solution and the second measurement was carried out 13 days after the injection of the Agrobacterium solution. The OJIP transients were measured using Handy PEA (Hansatech Instruments Ltd., Norfolk, UK). The determination result is based on three biological repetitions.

## Figures and Tables

**Figure 1 ijms-23-09887-f001:**
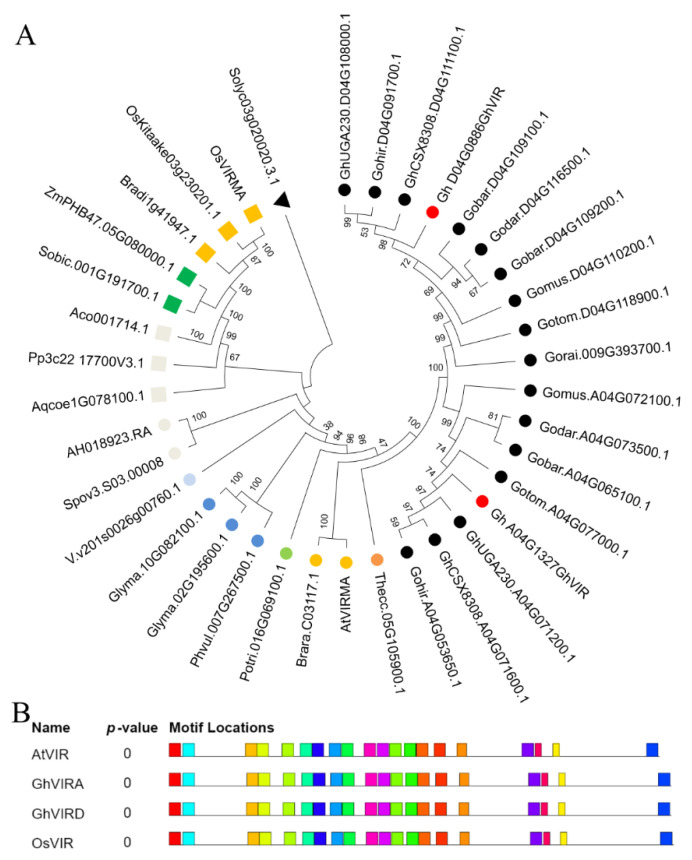
Polygenetic relationships of VIRMA homologs in different plant species. (**A**) Proteins from 37 different species (*Arabidopsis thaliana*, *Ananas comosus*, *Aquilegia coerulea*, *Amaranthus hypochondriacus*, *Brassica rapa*, *Brachypodium distachyon*, *Glycine max*, *Gossypium barbadense*, *Gossypium darwinii*, *Gossypium hirsutum CSX8308*, *Gossypium hirsutum UA48*, *Gossypium hirsutum UGA230*, *Gossypium hirsutum TM1*, *Gossypium mustelinum*, *Gossypium raimondii*, *Gossypium tomentosum*, *Oryza sativa*, *Populus trichocarpa*, *Phaseolus vulgaris*, *Physcomitrium patens*, *Spinacia oleracea*, *Sorghum bicolor*, *Solanum lycopersicum*, *Theobroma cacao*, *Vitis vinifera*, and *Zea mays*) are indicated by different icons and are classified into two clades. All available gene names are also indicated. The level of statistical support was conducted by the neighbor-joining method, and numbers on the major branches indicate bootstrap values. (**B**) Conversed motifs in GhVIR-A, GhVIR-D, AtVIR and OsVIR proteins.

**Figure 2 ijms-23-09887-f002:**
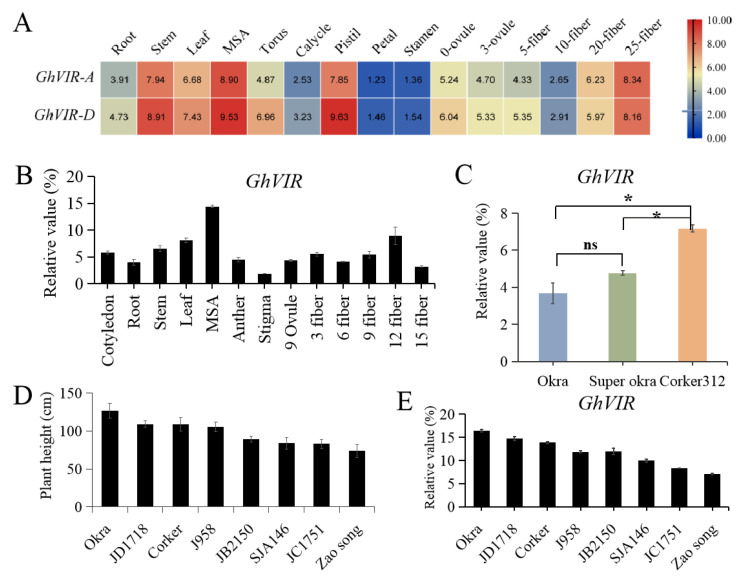
Expression patterns of *GhVIR-A/-D* in upland cotton. (**A**) Heat map analysis of *GhVIR-A/-D* genes expressions in different organs of upland cotton. The color from blue to red indicates low to high expression. (**B**) Quantitative real-time PCR (qRT-PCR) analysis of *GhVIRs* in various tissues of upland cotton. Results were normalized against the expression level of *GhUBI1*. MSA, main stem apex; 9 Ovule, Ovule at 9-day post-anthesis; 3–15 fiber, 3–15 day post anthesis fiber. Error bars indicate SD. (**C**) Quantitative real-time PCR analysis of *GhVIR* in the third leaf of cotton species with different leaf shape. (**D**) Comparison of plant height of eight allotetraploid cotton cultivars (*G. hirsutun*). The heights of plants were calculated when cotton bolls were opening. (**E**) QRT-PCR analysis expression of *GhVIR* in main stem apex (MSA) of different cotton species. The X-axis represents different upland cotton species, while the Y-axis represents gene relative expressions. Results were normalized against the expression level of *GhUBI1*. Error bars indicate SD of three biological replicates (* *p* < 0.05), ns = no significance.

**Figure 3 ijms-23-09887-f003:**
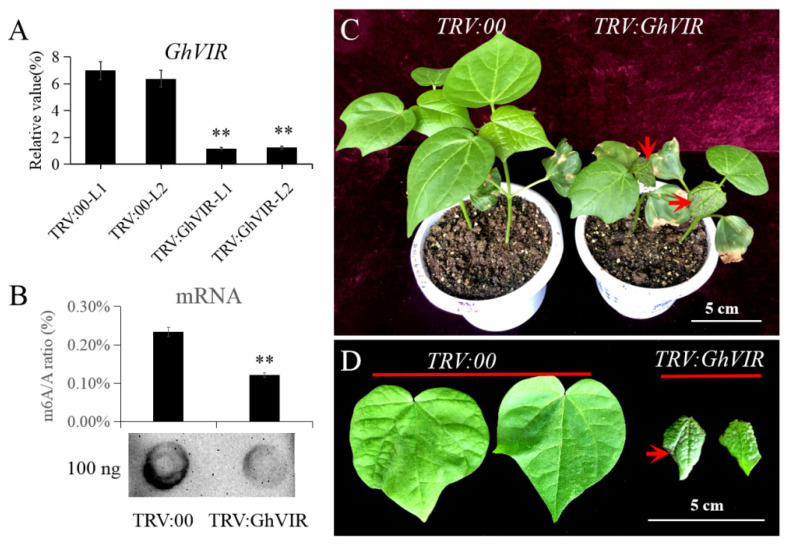
Phenotypic analysis of *TRV:GhVIR* plants. (**A**) Expression analysis of the second leaf of 2-leaf stage *TRV:GhVIR* plants and TRV:00 plants. (**B**) mRNA m^6^A level analysis of the second leaf of 2-leaf stage *TRV:GhVIR* plants and *TRV:00* plants. Up-penal: mRNA m^6^A/A ratio (%) of the second leaf of 2-leaf stage *TRV:GhVIR* plants and *TRV:00* plants by m^6^A RNA methylation quantification kit. Down-penal: dot blot analysis mRNA m^6^A level of the second leaf of 2-leaf stage *TRV:GhVIR* plants and *TRV:00* plants. Error bars indicate SD of three biological replicates (** *p* < 0.01) for (**A**,**B**). (**C**) Whole plant phenotype comparison of 3–4 leaf stage *TRV:GhVIR* plants with *TRV:00* plants (negative control group). (**D**) Second leaf phenotype comparison of 3–4 leaf stage *TRV:GhVIR* plants with *TRV:00* plants.

**Figure 4 ijms-23-09887-f004:**
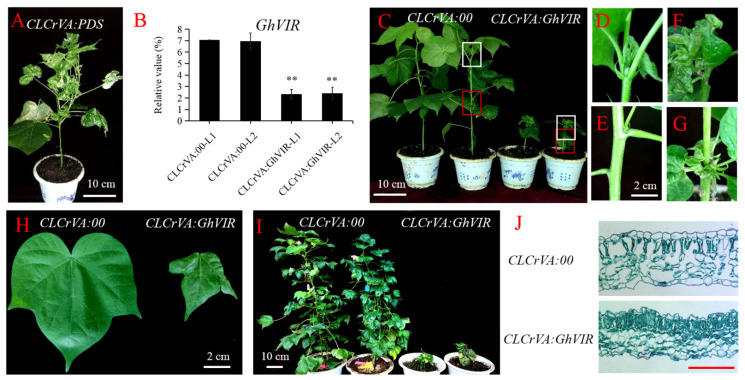
Functional characterization of GhVIR using VIGS (CLCrVA system). (**A**) Phenotypes of silencing *GhPDS* (positive control). (**B**) Expression analysis of the second leaf of 3-leaf stage *CLCrVA:GhVIR* plants and *CLCrVA:00* plants. Error bars indicate SD of three biological replicates (** *p* < 0.01). (**C**) Whole plant phenotype of 10 leaf stage *CLCrVA:00* plants and *CLCrVA:GhVIR* plants. (**D**,**E**) Magnified views of boxes in *CLCrVA:00* plants indicated in (**C**), respectively. (**F**,**G**) Magnified views of boxes in *CLCrVA:GhVIR* plants indicated in (**C**), respectively. (**H**) Third leaf phenotype comparison of 5-leaf stage *CLCrVA:GhVIR* plants with *CLCrVA:00* plants. (**I**) Whole plant phenotype of blooming stage of *CLCrVA:00* plants and *CLCrVA:GhVIR* plants. (**J**) Longitudinal sections of leaf blade of *CLCrVA:00* plants and *CLCrVA:GhVIR* plants.

**Figure 5 ijms-23-09887-f005:**
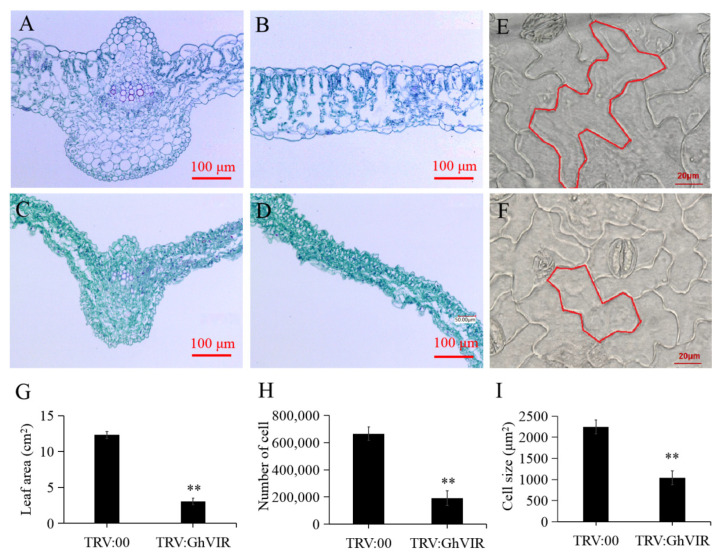
Knock-down *GhVIRs* affect leaf morphogenesis during cotton leaf proliferation and expansion developmental stage. (**A**–**D**) Longitudinal sections of paraffin-embedded leaf were stained with toluidine blue (**A**–**D**) to show cell morphology of leaf vein and leaf blade. (**A**,**B**) longitudinal sections of leaf vein and leaf blade of *TRV:00* plants, respectively. (**C**,**D**) longitudinal sections of leaf vein and leaf blade of *TRV:GhVIR* plants, respectively. (**E**,**F**) Super depth of field microscope pictures of the abaxial epidermal cells of the second leaf of *TRV:00* and *TRV:GhTRV* plants. Bar = 20 μm. (**G**–**I**) The leaf area, total cell number, and cell size of the second leaf of *TRV:00* and *TRV:GhTRV* plants. Error bars indicate SD of three biological replicates (** *p* < 0.01).

**Figure 6 ijms-23-09887-f006:**
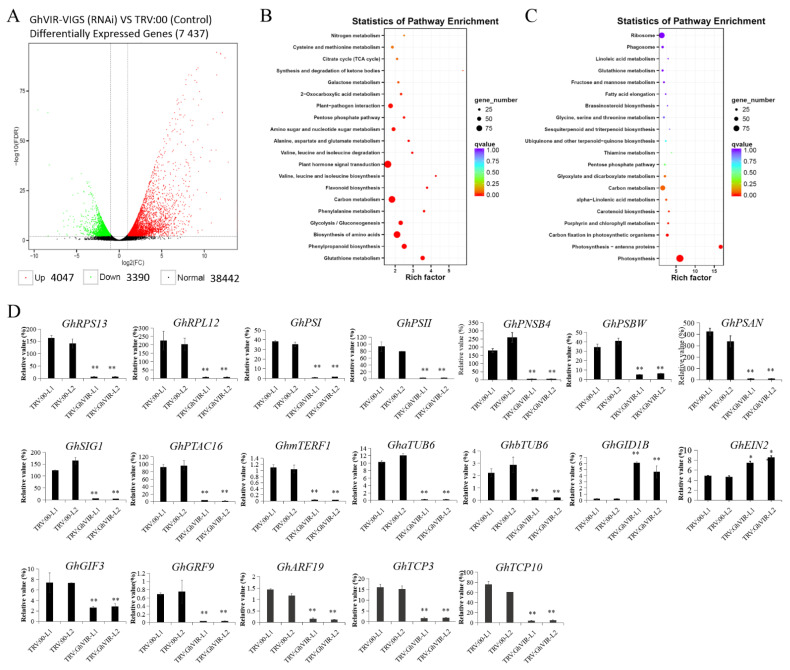
RNA-Seq analysis of the differentially expressed genes (DEGs) in the second leaf of 2-leaf stage cotton of *TRV:GhVIR* plants. (**A**). Volcano of DEGs between *TRV:00* and *TRV:GhVIR* plants. Red dots (upregulated) and green dots (down-regulated) indicate the up- and down-regulated genes in *TRV:GhVIR* leaves, respectively, compared with the *TRV:00* leaves. (**B**). TRV:GhVIR vs. TRV:00 up. DEGs enriched KEGG pathway scatterplot. (**C**). TRV:GhVIR vs. TRV:00 down.DEGs enriched KEGG pathway scatterplot. (**D**). qRT-PCR analysis of genes related to leaf development in the second leaf of *TRV:GhVIR* plants. Total RNA was isolated from the second leaf of *TRV:GhVIR* plants and *TRV:00* plants. *GhUBI1* (EU604080) was used as internal reference. Error bars represent SD of three biological replicates. * *p* < 0.05; ** *p* < 0.01 by independent *t*-tests.

**Figure 7 ijms-23-09887-f007:**
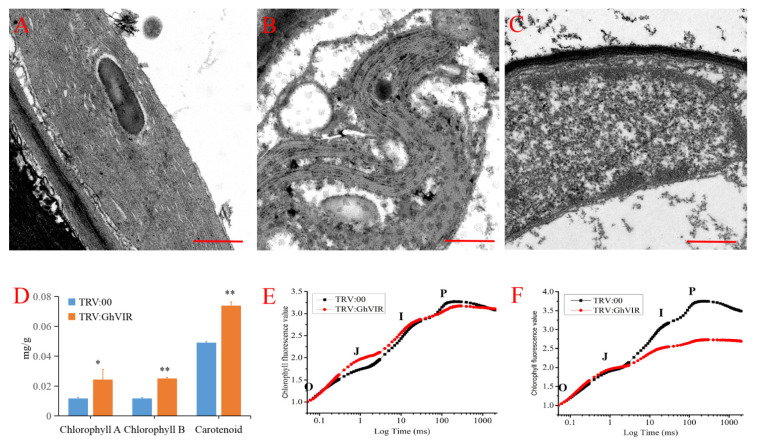
Chloroplast ultrastructure, pigment content, and photosynthetic efficiency analysis in cotton leaves of *TRV:00* plants and *TRV:GhVIR* plants. (**A**,**C**) Chloroplast ultrastructures of the *TRV:00* plant (**A**), *TRV:GhVIR-1* plant (**B**), and *TRV:GhVIR-2* plant (**C**). Bar = 500 nm. (**D**) Photosynthetic pigment content in *TRV:00* plant and *TRV:GhVIR* plants. (**E**,**F**) Detection of photosynthetic efficiency of cotton leaves in *TRV:00* plant and *TRV:GhVIR* plants by OJIP transients. (**E**) Cotton leaves 10 d after injection of the *Agrobacterium* solution, and (**F**) cotton leaves 13 d after injection of the *Agrobacterium* solution. Error bars represent SD of three biological replicates. * *p* < 0.05; ** *p* < 0.01 by independent *t*-tests.

**Table 1 ijms-23-09887-t001:** VIR genes in upland cotton (*Gossypium hirsutum* L. acc. TM-1) ^a^.

Gene Name ^b^	Gene Symbol	Length (a.a.)	MW (Da)	pI	Ortholog Gene Name and ID in *Arabidopsis*	Ortholog Gene Name and ID in *Oryza sativa*
GhVIR-A	Gh_A04G1327	2189	239.5	5.32	AtVIR AT3G05680 (2138 aa)	OsVIR OsKitaake03g230201.1 (2199 aa)
GhVIR-D	Gh_D04G0886	2188	239.7	5.25		

^a^ Gene information in *G. hirsutum* from Zhang et al. (2015). ^b^ A and D were derived from the A-subgenome and D-subgenome progenitor in the tetraploid cotton. The *Gene symbol* was named by Nanjing Agricultural University, Nanjing, China and sequenced by Novogene Bioinformatics Institute (NBI), Beijing, China.

## Data Availability

The original data presented in the study are included in the article and Appendix A. Further inquiries can be directed to the corresponding authors.

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
