# Peer review of "Cotton (Gossypium hirsutum) VIRMA as an N6-Methyladenosine RNA Methylation Regulator Participates in Controlling Chloroplast-Dependent and Independent Leaf Development"

_ijms, 2022, doi:10.3390/ijms23179887_

Round 1
Reviewer 1 Report
Dear Authors,
The study seems to be valuable. It has cognitive merit and the research results are quite well documented. The manuscript is well written.
However, there are a few points of concern, supplementations and corrections (attached pdf file).

Author Response
Reviewer #1
Comments:
The “Introduction” is very informative and at the end contains the research goals. Moreover, is well written. The Results are well written, but some parts are more suitable to other sections. The “Discussion” section is sufficiently carefully analyzed. The Materials and methods section is well described but some parts from the Results must be moved to this one.
- Introduction section (Line49-55): Please add more information from:
https://doi.org/10.1038/nchembio.1432, https://doi.org/10.1038/ncb2902, https://doi.org/10.1093/nar/gkz619 and https://doi.org/10.1038/cr.2014.3.
Reply 1: According to your kind suggestion, we have add more information to introduce the basis information of mRNA m6A methyltransferase complex (Page 2, lines 50-59, 68), and increased the new references (new references No. 10-18). Thank you very much.
- Results section: It is described in Materials and methods, please rephrase.
Reply 2: According to your kind suggestions, we have rephrased some sentences in “Results” and described these informations to “Materials and methods” section, for example, Results--2.1 and 2.2 (page3 lines 125-127; page4 137; page5 176, 182).
And other sentences can increase the coherence and logic of the results section, we think it can be presented in the “Results” parts, but not in the “Materials and Methods” or “Discussion”. for example, Results--2.3 and 2.4. Thank you very much.
- Materials and methodssection: The section is well described but some parts from the Results must be moved to this one.
Reply 3: According to your kind suggestions, we have rephrased some sentences in “Results” and described these informations to “Materials and methods” section, for example, Results--2.1 and 2.2. Thank you very much.
The revised word version of this manuscript was showed in upload file.
Reviewer 2 Report
the research framework is very interesting and the paper is well written. the introduction gives enough information about the research topic. materials and methods are informative and make it easy to read and repeat similar experiments. the results are clearly presented and discussed. the conclusion is based on the presented results. however, I have questions about lines 493- 495 on page 13 the authors said that GhVIRs regulate 494 the leaf and MSA development, and affect the cotton architecture. is it possible to explain it more for example what are the signs and expression of that influence, besides changing the structure of leaves.
Author Response
Reviewer #2
Comments:
The research framework is very interesting and the paper is well written. the Introduction gives enough information about the research topic. Materials and Methods are informative and make it easy to read and repeat similar experiments. the Results are clearly presented and discussed. the Conclusion is based on the presented results. However, I have questions about lines 493-495 on page 13 the authors said that GhVIRs regulate the leaf and MSA development, and affect the cotton architecture. Is it possible to explain it more for example what are the signs and expression of that influence, besides changing the structure of leaves.
Reply 1: The “Result” of GhVIRs affect the cotton architecture was showed in page 7 lines 251-254. And according to your kind suggestion, we have rephrased the “Conclusion” in lines 493- 495. Thank you very much.
The revised word version of this manuscript was showed in upload file.